# Preliminary characterisation of the spatial immune and vascular environment in triple negative basal breast carcinomas using multiplex fluorescent immunohistochemistry

**Elena A. Takano**[1]*, **Metta K. Jana**[2], **Luis E. Lara Gonzalez**[3,4], **Jia-Min B. Pang**[1], **Roberto Salgado**[3,5], **Sherene Loi**[3,4], **Stephen B. Fox**[1,4]*

1 Department of Pathology, Peter MacCallum Cancer Centre, Melbourne, Victoria, Australia, 2 Centre for Advanced Histology and Microscopy, Peter MacCallum Cancer Centre, Melbourne, Victoria, Australia, 3 Cancer Research Division, Peter MacCallum Cancer Centre, Melbourne, Victoria, Australia, 4 The Sir Peter MacCallum Department of Oncology, The University of Melbourne, Parkville, Victoria, Australia, 5 GZA-ZNA-Hospitals, Antwerp, Belgium

* elena.takano@petermac.org (EAT); stephen.fox@petermac.org (SBF)

**Data Availability Statement:** All relevant data are within the manuscript and its Supporting Information files.

## Abstract

Triple negative breast cancers often contain higher numbers of tumour-infiltrating lymphocytes compared with other breast cancer subtypes, with their number correlating with prolonged survival. Since little is known about tumour-infiltrating lymphocyte trafficking in triple negative breast cancers, we investigated the relationship between tumour-infiltrating lymphocytes and the vascular compartment to better understand the immune tumour microenvironment in this aggressive cancer type. We aimed to identify mechanisms and signaling pathways responsible for immune cell trafficking in triple negative breast cancers, specifically of basal type, that could potentially be manipulated to change such tumours from immune "cold" to "hot" thereby increasing the likelihood of successful immunotherapy in this challenging patient population. We characterised the spatial immune environment in 10 basal breast cancers showing a range of tumour-infiltrating lymphocytes using multiplex fluorescent immunohistochemistry and quantitative digital analysis of CD3$^+$ T cells. We examined their relationship to blood vessels and their activation status as defined by VCAM-1, ICAM-1 and PD-L1. Confirmation of the relationship between tumour-infiltrating lymphocytes and endothelial activation was performed through *in silico* analysis on TCGA BRCA RNA-seq data (N = 808). Significantly higher CD3$^+$ T cell densities were observed in the stromal compartment compared with the neoplastic cell compartment (P = 0.003). ICAM-1 activated blood vessels were spatially associated with higher CD3$^+$ T cell densities only within 30 microns of blood vessels compared with more distal activated and non-activated blood vessels (P = 0.041). *In silico* analysis confirmed higher numbers of tumour-infiltrating lymphocytes in basal breast cancers and that higher numbers were significantly associated with endothelial cell activation molecules, co-clustering with upregulated *ICAM-1 and VCAM-1* amongst others. PD-L1 was also identified in a subset of blood vessels, suggesting an additional immune regulatory mechanism in endothelial cells. Regulating the activation

**Funding:** SBF is supported by NHMRC Investigator grant GNT1193630. (https://www.nhmrc.gov.au/funding/find-funding/investigator-grants).

**Competing interests:** The authors have declared that no competing interests exist.

status of tumour-associated vascular endothelial cells may improve T cell trafficking into basal breast tumours and enhance immunotherapeutic response.

## Introduction

Triple negative breast cancers (TNBC) is a breast carcinoma characterised by the absence of estrogen receptors (ER), progesterone receptors (PR) and human epidermal growth factor receptor 2 (HER2) as defined by protein and/or gene amplification [1]. TNBC is a heterogeneous group of tumours that can be classified further into subtypes based on their morphology and transcriptomic profile [2]. The most common is basal-like with other groups being claudin-low, mesenchymal, luminal androgen receptor and immunomodulatory. Basal-like breast cancers account for approximately 15% of all breast cancers and are often seen in patients harbouring a BRCA1 mutation. They are genomically unstable, exhibit high histologic grade and are generally refractory to treatment [3]. The absence of the usual therapeutic targets from their triple negative phenotype has led to the exploration of other treatment modalities, including immunotherapy [4]. This approach is particularly attractive in these tumours as evident from the frequent presence of tumour-infiltrating lymphocytes (TILs), even in early-stage tumours. The number of TILs is associated with improved prognosis and response to chemotherapy in both adjuvant and neoadjuvant contexts [5–12]. Furthermore, expression of the checkpoint inhibitor programmed cell death ligand 1 (PD-L1) in TNBCs is primarily found on tumor-infiltrating immune cells and is correlated with longer overall survival, distinguishing it from lung cancers where it is expressed in tumor cells [13]. Recently, it has been shown that patients with PD-L1 expression treated with the checkpoint inhibitor pembroluzimab and atezolizumab have significantly longer survival [13, 14], although the confirmatory study with atezolizumab failed to show a significant increase in survival [15], raising questions about the role of PD-L1 a predictive marker. PD-L1 in TNBCs continues to be a subject of ongoing investigation.

Nevertheless, not all TNBCs of basal type have high numbers of TILs, with many tumours having none or low numbers of TILs; and even those with high TILs often display a variable PD-L1 activation status. Naturally, one of the processes that the host must establish for an effective immune response is to traffic TILs into the tumour site. Understanding how tumours modulate the immune response is increasingly important for determining the nature of the tumour microenvironment [16], particularly the status of blood vessels (BVs), which not only supply the nutrients to the growing tumour but also serve as conduits for immune cells as they enter and exit the tumour [17]. In our previous work, we demonstrated that platelet cell adhesion molecule 1 (PECAM-1), E and P selectins along with intercellular adhesion molecule 1 (ICAM-1), members of the immunoglobulin superfamily that mediate cancer metastasis and leukocyte migration [17], are upregulated in breast cancer associated endothelium, especially at the tumour periphery. Others have shown that vascular cell adhesion molecule 1 (VCAM-1), facilitating trans-endothelial migration is expressed on both endothelium and tumour cells allowing interaction with very late antigen-4 (VLA-4) antigens on leukocytes to activate PI3K pathway for tumour progression [18]. These studies support the significant role of adhesion molecules on breast tumour-associated vascular endothelium in the immune cell trafficking and the immune response [17].

While various aspects of the tumour microenvironment including angiogenesis and immune system have been previously studied, these have been investigated independently and

in different contexts. The objective of this study was to examine the relationship between the activation status of blood vasculatures and immune infiltrates in the context of basal type TNBC. Specifically, we aimed to characterise T cells, including any preferential spatial location in relation to the vasculature and the activation status of the endothelium. Additionally, we sought to examine the expression of PD-L1 in epithelial, endothelial and stromal component of TNBCs of basal type. The overarching goals were to gain a better understanding of the process of TILs trafficking into this tumour type and to identify potential targets that could be modulated to improve the likelihood of responses to immune therapies.

## Materials and methods

### Sample cohort

Ten basal TNBC were selected from the archival formalin-fixed paraffin-embedded (FFPE) collection at Peter MacCallum Cancer Centre by accessing the laboratory information management systems in September 2018. The FFPE blocks were accessed for IHC on 23/10/2018. The study was conducted in compliance with the National Statement on Ethical Conduct in Human Research and the approval was obtained from Human Research Ethics Committee at Peter MacCallum Cancer Centre (Ethics 10/16). Patients had either given broad written consent to future research with their samples and data, or waivers of consent were in place. The median age of the patients at surgery was 53.5 (range 27 to 67). All tumours were grade 3 invasive carcinomas of no special type that were negative for ER, PR and HER2 on immunohistochemistry and non-amplified by *in situ* hybridization. All tumours were of a basal-like phenotype with positivity for EGFR and/or CK5/6 immunohistochemistry. The cases were selected to contain a range of TILs (range 5% - 90%, median 65%) as assessed by the method of the International Immuno-Oncology Biomarker Working group [19].

### Immunohistochemistry

To determine the spatial environment of basal TNBCs, whole sections from 10 cases were examined by multiplex fluorescent immunohistochemistry (mIF) using the Opal™ 7-Colour Manual IHC Kit (PerkinElmer, USA) and antibodies to CD3 (Spring Bioscience, M3074), PD-L1 (Ventana, 790–4905), ICAM-1 (ThermoFisher, MA5-11433), pan-cytokeratin (Novocastra, NCL-L-AE1/AE3), VCAM-1 (ThermoFisher, MA5-11447) and CD31 (Dako, M0823) (Table 1 and supporting information for detailed methods, including optimised staining conditions).

### Image acquisition and segmentation of tumour compartments

Imaging of the mIF slides was performed on the quantitative Pathology Imaging System Vectra 3 (PerkinElmer) with exposure time optimised for each channel. A low-resolution image (10x) of the whole tissue scan was first taken before the selection of the regions of interests (ROI) for high resolution acquisition with 20x objective lens. The ROIs were selected using a whole slide image viewer, Phenochart (Akoya Biosciences). Areas from the tumour periphery which is recognized to be most biologically important [20] and where the cells of our interest such as immune cells, microvasculature endothelial cells and tumour cells are most prominently located. The single colour control slides, each stained with a single antibody were used to build spectral libraries for spectral unmixing in Inform 2.4.1 software (PerkinElmer, USA) which also served as a control for the staining specificity. The captured images were then analysed in HALO software (Indica Labs, USA) for tissue and cell segmentations to yield the quantitative data (S1 Fig). The cells were phenotyped using positive expression of the different TSA

**Table 1. Optimised antibody staining order and conditions used to stain the basal breast cases.**

| Order | Retrieval buffer | Primary ab (clone) | Primary ab dilution | Primary ab incubation condition | Polymer HRP Secondary ab (Dako), 30mins RT | TSA fluorophore (PerkinElmer) at 1/100, 10mins RT |
|---|---|---|---|---|---|---|
| 1 | AR6 | CD3 (SP7, Rabbit) | 1/500 | 60mins RT | EnvisionFLEX HRP | Opal 650 |
| 2 | pH8 | PD-L1 (SP263, Rabbit) | Neat | 60mins RT | EnvisionFLEX HRP | Opal 540 |
| 3 | pH8 | ICAM-1 (23G12, Mouse) | 1/10 | 60mins RT | EnvisionFLEX HRP | Opal 620 |
| 4 | AR6 | pan-Cytokeratin (AE1/AE3, Mouse) | 1/200 | 60mins RT | EnvisionFLEX HRP | Opal 690 |
| 5 | pH8 | VCAM-1 (1.4C3, Mouse) | 1/50 | o/n at 4°C | EnvisionFLEX HRP | Opal 520 |
| 6 | pH8 | CD31 (JC/70A, Mouse) | 1/400 | 60mins RT | EnvisionFLEX HRP | Opal 570 |

fluorophores and for segmentation prepared for the downstream analysis including the spatial relationships between tumour cells, immune cells and the blood vasculature (see supporting information and S2 Fig).

## Quantification and spatial analysis of immune, endothelial cells and blood vessels

Quantification of cells of various types and phenotypes was performed in HALO. The cells with DAPI positive nuclei defined the total number of cells. The markers pan-cytokeratin, CD3 and CD31 defined tumour cells, T cells and endothelial cells (ECs) respectively, and the other markers PD-L1, ICAM-1 and VCAM-1 identified the activated phenotypes of these cells. HALO software tabulated the cell number of each type based on the marker positivity. Cell density was calculated from the number of cells present in each sample in the tumour and stromal compartments and expressed as the number of cells±marker positive/mm$^2$.

Vessel-like structures made up of ECs expressing CD31 were defined as BVs. BV density was calculated from the number of BVs in each sample and quantitated using HALO AI software (Indica Labs, USA). Briefly, the DenseNet V2 network algorithm was selected to train and build analysis classifiers which define each vessel to a phenotype. Only the vessels with CD31$^+$ expression, defining BVs, were quantified and each BV was assigned a phenotype accordingly to its expression. The "positivity" was assigned to the vessels when whole or parts of vessel structure expressed positive signals. The number of vessels in each phenotype was then expressed as BV density (BVs/mm$^2$) (see S1 Table as an example).

The spatial relationship of immune cell infiltration to blood vessels was carried out using the HALO Spatial Analysis module, which measures the CD3$^+$ density (cells/mm$^2$) up to 120μm from each blood vessel, phenotyped according to their VCAM-1, ICAM-1 or PD-L1 activation status (S3 Fig).

## *In Silico* analysis of tumour microenvironment on The Cancer Genome Atlas (TCGA) breast cancer data

I*n silico* analysis of breast cancer cases from TCGA was performed additionally to confirm and to further explore the relationship between TILs and endothelial cells and their activation status. For this analysis, we used RNA-seq data downloaded from the Xena browser [21] (v.07-18-2019), which consisted of $log_2$ transformed HTSeq counts [22] recommended for single cohort comparison. The final dataset for the unsupervised clustering analysis included the cases with available PAM50 subtype, also downloaded from the Xena browser, and TIL measurements made by a pathologist and a machine learning method [23], leading to a total of 808

cases. They comprised of basal (N = 147), luminal A (N = 415), luminal B (N = 176), and HER2-positive (HER2+, N = 70).

RNA-seq data were reversed to counts as $2^{x}-1$, where $x$ is the transformed value, to be normalised by the variance-stabilising transformation vst from the DESeq2 [24] R package, v1.34.0. The normalised data were then standardised using the code of the function coolmap from the limma package v3.50.3. A gene set, comprising vascular endothelium and activation adhesion molecules of interest, along with a recognised gene profile for TILs, was utilised for clustering [25–27]. We performed unsupervised hierarchical clustering with Euclidean distance and Ward's minimum variance as the agglomeration method on standardised values. Four clusters were identified by visual inspection. These were validated by evaluating the CD3D expression values, which were labelled as follows:

- Downregulated if the median expression was below 0.

- Expressed if the median expression value is above 0.

- Upregulated if the median expression value is above 1.

Once the clusters were labelled, a $\chi^2$ test for independence between cluster labels and breast cancer subtype was evaluated. Results were displayed as the percentage of the contribution using the Pearson residuals ($r^2$), calculated as $contribution = \frac{r^2}{\chi^2}$, using the corrplot R package v.0.92.

Differential expression was performed on reversed counts as described above, using DESeq2 (d). Genes with low counts were removed if they had fewer than 3 reads in total, leaving 16,570 genes for the analysis. P-values were adjusted using the false discovery rate, with a significance threshold of $\alpha$ = 0.01.

## Statistical analysis

The significance level used in this study was α = 0.05. Unless otherwise specified in each figure, all statistical tests between groups were determined by unpaired, two-tailed t-test with Welch's correction using GraphPad Prism v.10. Unless otherwise specified, statistical testing in the *in silico* analyses were unpaired, two-tailed Wilcoxon rank-sum tests. P values were adjusted for multiple comparisons using the Benjamin-Hochberg method. RNA-sequencing analysis was performed in R v.4.1.3.

## Results

### CD3+ immune cells in TNBC reside in the stromal rather than the tumour compartment

CD3+ T cells were present in both the tumour and stromal compartments (S4 Fig). The overall tumour (neoplastic and non-neoplastic) median CD3+ T cell density (cells/mm$^2$) was 1409.8 cells/mm$^2$ (range 636.0–3607.4 cells/mm$^2$) with 232.1 cells/mm$^2$ (range 12.7–2419.8 cells/mm$^2$) in the tumour compartment, and 1642.5 cells/mm$^2$ (range 741.1–4116.2 cells/mm$^2$) in the stromal compartment. There were significantly more CD3+ T cells in the stromal than in the neoplastic compartment (P = 0.003, Fig 1A).

### A subset of tumour-associated CD3+ immune cells in basal TNBC express the activation markers VCAM-1, ICAM-1 or PD-L1

Examples of tumour-associated CD3+T cells expressing various activation markers are shown in Fig 2.

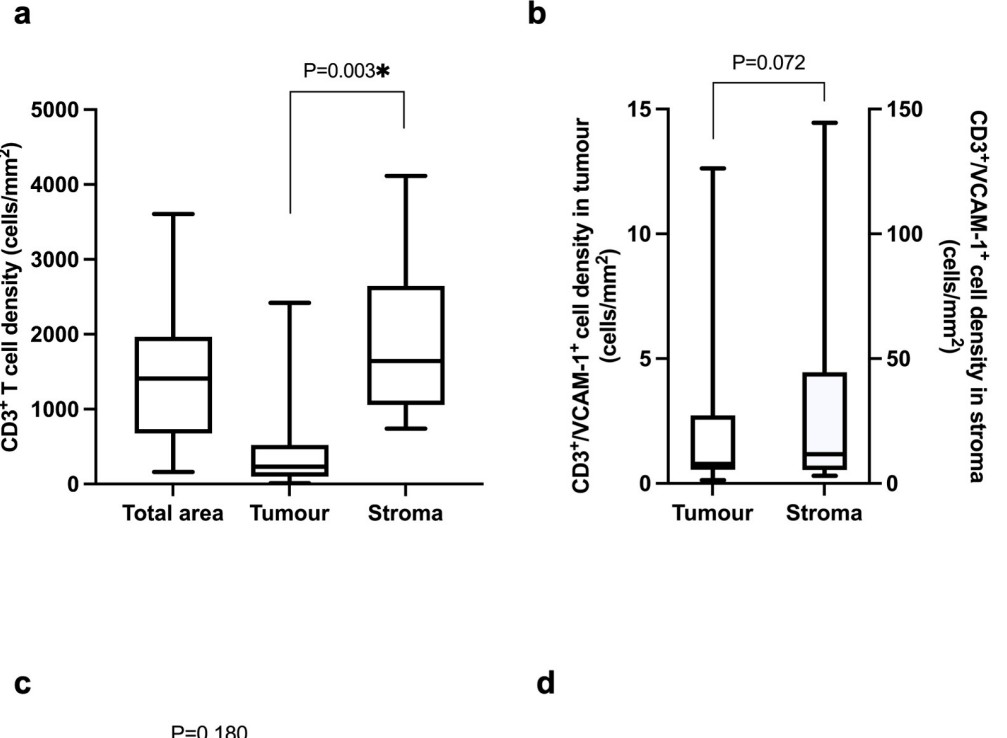

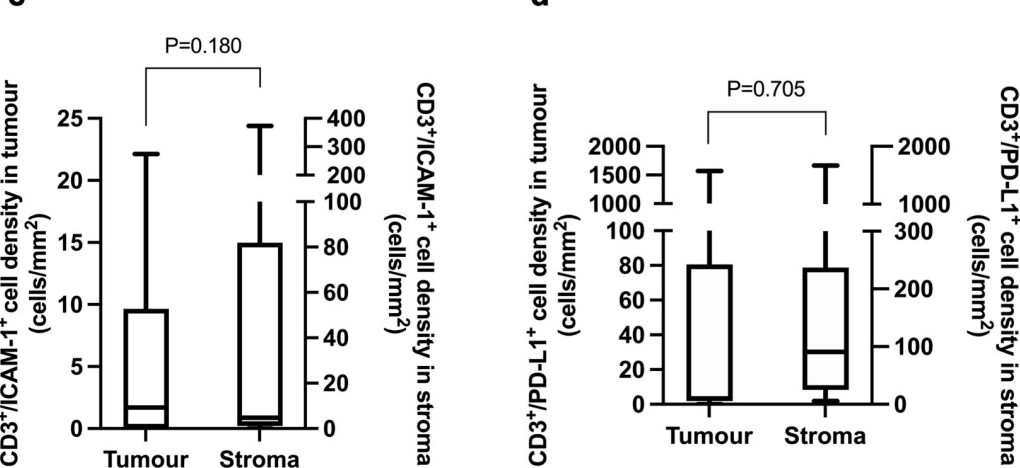

**Fig 1. Characterisation of CD3⁺ T cell density in basal TNBCs. a)** CD3⁺ T cells in total tumour, neoplastic cell and stromal compartments showing significantly higher numbers in stromal compared with tumour cell compartment. **b-d)** sub-populations of CD3⁺ T cells densities in each compartment co-expressing the activation markers **b)** VCAM-1, **c)** ICAM-1 and **d)** PD-L1 showing no significant difference between tumour and stromal compartments.

VCAM-1 was expressed in 0.9% of CD3⁺ T cells overall, and in tumour and stromal compartments in 0.4% (range 0.1–7.4%) and 1.0% (range 0.1–8.1%) of CD3⁺ T cells respectively. The overall median CD3⁺/VCAM-1⁺ T cell density was 8.1 cells/mm² (range 2.7–66.0 cells/mm²) with non-statistically significant higher numbers of CD3⁺/VCAM-1⁺ T cells in the stromal compartment (11.7 cells/mm², range 3.1–144.5 cells/mm²) compared with the tumour compartment (0.8 cells/mm², range 0.1–12.6 cells/mm², Fig 1B).

ICAM-1 was expressed in 0.3% of CD3⁺ T cells overall, and in tumour and stromal compartments in 0.5% (range 0.0–61.1%) and 0.3% (range 0.0–50.4%) of CD3⁺ T cells respectively. The overall median CD3⁺/ICAM-1⁺ T cell density was 5.0 cells/mm² (range 0.1–144.9 cells/mm²), with non-statistically significant higher numbers of CD3⁺/ICAM-1⁺ T cells in the

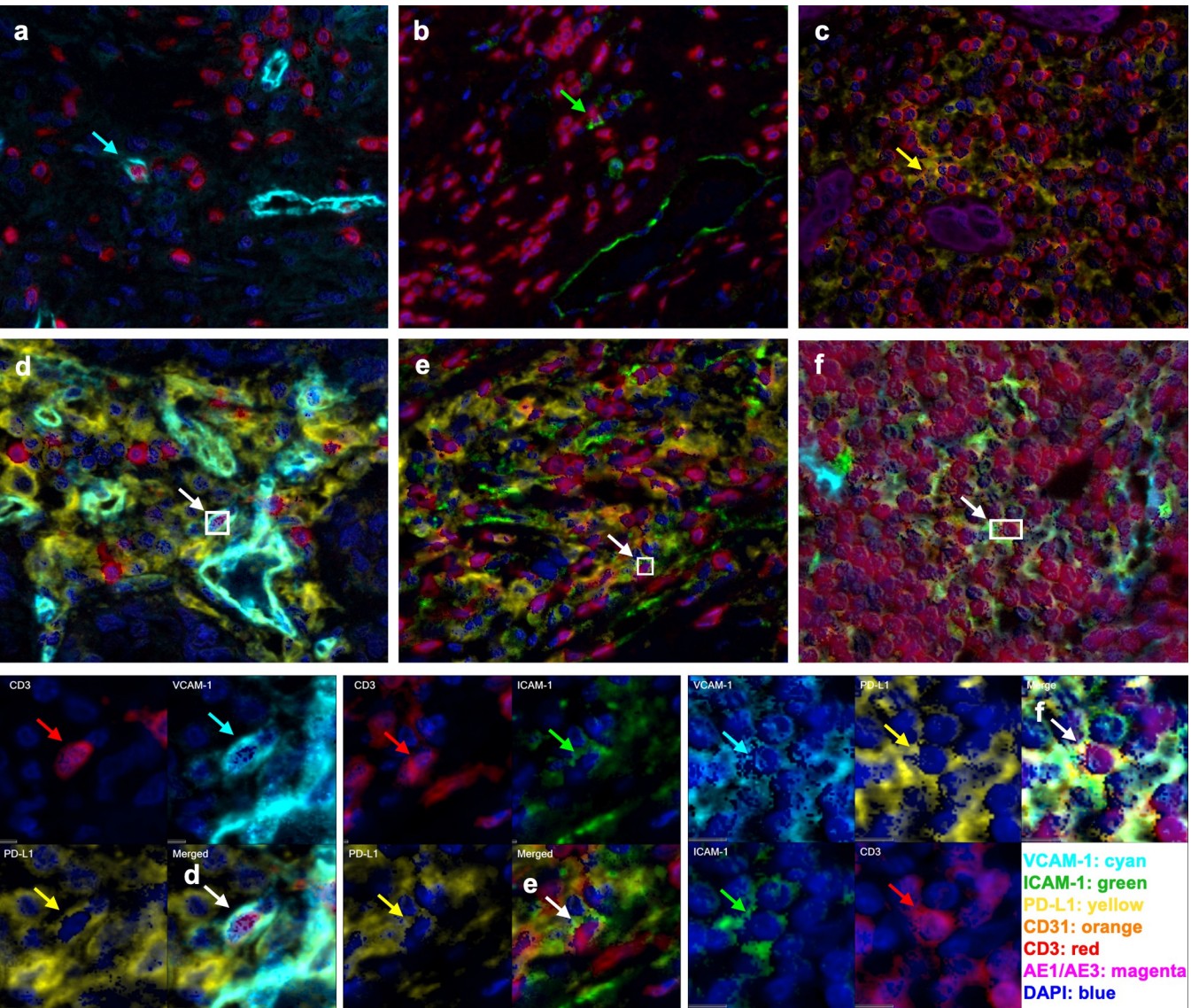

**Fig 2. Activation markers on CD3$^+$ immune cells in basal TNBC.** A subset of CD3$^+$ T cells expressing **a)** VCAM-1, **b)** ICAM-1, **c)** PD-L1, **d)** VCAM-1 and PD-L1, **e)** ICAM-1 and PD-L1 and **f)** VCAM-1, ICAM-1 and PD-L1. Examples of each phenotype indicated by arrows of their respective colours. The montages in the last row display a single channel with DAPI for each of the corresponding panel d), e) and f).

stroma (4.8 cells/mm$^2$, range 0.1–373.3 cells/mm$^2$) than in the tumour compartment (1.7 cells/mm$^2$, range 0.0–22.1 cells/mm$^2$, Fig 1C).

PD-L1 was expressed in 7.1% of CD3$^+$ T cells overall; and by 3.8% (range 0.0–64.9%) and 7.5% (range 0.3–44.7%) of CD3$^+$ T cells in tumour and stromal compartments respectively. The median CD3$^+$/PD-L1$^+$ T cell density was 39.6 cells/mm$^2$ (range 4.8–1658.1 cells/mm$^2$), with non-statistically significant higher numbers of CD3$^+$/PD-L1$^+$ T cells in the stroma (91.0 cells/mm$^2$, range 5.7–1667.0 cells/mm$^2$) compared with the neoplastic compartment (3.4 cells/mm$^2$, range 0.0–1570.6 cells/mm$^2$, Fig 1D).

Although rare, there were occasional cells that showed CD3$^+$/PD-L1$^+$/VCAM-1$^+$, CD3$^+$/PD-L1$^+$/ICAM-1$^+$ and CD3$^+$/PD-L1$^+$/VCAM-1$^+$/ICAM-1$^+$ (Fig 2D–2F). The median density

and proportion of each of the types are all below 1 cell/mm$^2$ and made up <0.05% of total CD3$^+$ T cells.

## A subset of ECs and BVs in basal TNBC express the activation markers PD-L1, VCAM-1 and ICAM-1

Examples of various BV phenotypes are shown in Fig 3A. Both EC and BV density identified by positive CD31 expression were quantified in the whole tumour (whole ROI).

The overall median tumour CD31$^+$ EC density was 131.4 cells/mm$^2$ (range 4.0–346.9 cells/mm$^2$, Fig 3B). The median proportion of non-activated ECs was 56.6% (range 19.2–82.1%) and the median proportion of activated ECs was 43.4% (range 18.4–80.8%). There was no significant difference between the median non-activated (69.9 cells/mm$^2$, range 0.8–183.8 cells/mm$^2$) and the median activated EC density (45.3 cells/mm$^2$, range 3.2–163.0 cells/ mm$^2$) (P = 0.295). The median proportion of PD-L1, VCAM-1 and ICAM-1 positive CD31$^+$ ECs was 7.1% (range 2.6–21.6%), 12.3% (range 4.7–42.3%) and 15.8% (range 0.4–53.7%) respectively. The median density of each of the activated ECs phenotype, PD-L1$^+$ (median 6.5 cells/mm$^2$, range 0.3–52.9 cells/mm$^2$), VCAM-1$^+$ (median 15.6 cells/mm$^2$, range 1.7–59.7 cells/mm$^2$) and ICAM-1$^+$ (median 8.8 cells/mm$^2$ (range 0.9–50.4 cells/mm$^2$) were significantly lower than median density of non-activated ECs (P = 0.017, P = 0.018, P = 0.017 respectively) (Fig 3B).

When assessing the vasculature by BV density as opposed to EC density, the overall median tumour BV density in the overall tumour was 53.6 vessels/mm$^2$ (range 20.3–88.6 vessels/mm$^2$, Fig 3C). The median proportion of activated BVs (82.0%, range 58.0–94.6%) was higher than the median proportion of non-activated BVs (15.1%, range 0–40.7%), with the total activated BV density (median 41.2 vessels/mm$^2$, range 19.2–64.8 vessels/mm$^2$) being significantly higher than the non-activated BV density (median 4.5 vessels/mm$^2$, range 0–27.4 vessels/mm$^2$) (P = 0.001). The density of BVs that were positive for either PD-L1, VCAM-1, ICAM-1 or double positive for VCAM-1 and ICAM-1 was 1.1 vessels/mm$^2$ (range 0–14.8 vessels/mm$^2$), 24.5 vessels/mm$^2$ (range 11.3–64.0 vessels/mm$^2$), 1.3 vessels/mm$^2$ (range 0–8.2 vessels/mm$^2$) and 4.4 vessels/mm$^2$ (range 0–24.3 vessels/mm$^2$) respectively (Fig 3C). Thus, there were significantly higher density of VCAM-1 positive BV when compared with non-activated BVs (P = 0.014). However, although not significant, a lower density of ICAM-1 positive BV was observed when compared with non-activated BVs (P = 0.060) (Fig 3C). Furthermore, there was a significant increase in the density of VCAM-1$^+$ activated BVs than both ICAM-1$^+$ (P = 0.002) and PD-L1$^+$ activated BVs (P = 0.002) (Fig 3C).

## Increased CD3$^+$ T cell densities are identified proximally rather than distally to activated or compared with non-activated BVs

There was no significant difference between overall T cell densities around activated BVs when combined (i.e. VCAM-1$^+$, ICAM-1$^+$, VCAM-1$^+$/ICAM-1$^+$) within 0-30μm (median 2612.9 cells/mm$^2$, range 244.2–5689.4 cells/mm$^2$) compared with non-activated BVs (median 1894.5 cells/mm$^2$, range 141.3–4490.9 cells/mm$^2$) (P = 0.374) (Fig 4A). However, a significant increase in CD3$^+$ T cell density was specifically observed around ICAM-1 activated BVs compared with non-activated BVs or VCAM-1 activated BVs within the most proximal 30μm zone (P = 0.041, P = 0.009 respectively) (Fig 4B). Conversely, no significant difference in CD3$^+$ T cell densities was observed around VCAM-1$^+$ BVs compared with non-activated BVs (P = 0.489) (Fig 4B) in the 30μm zone. A significant reduction in CD3$^+$ T cell density was observed when moving from proximal to distal in ICAM-1 (when compared between the most proximal 30μm zone compared with the most distal 30μm zone at 90μm-120μm, P = 0.031) but not VCAM-1(P = 0.320), doubly expressing VCAM-1/ICAM-1 activated BVs (P = 0.078)

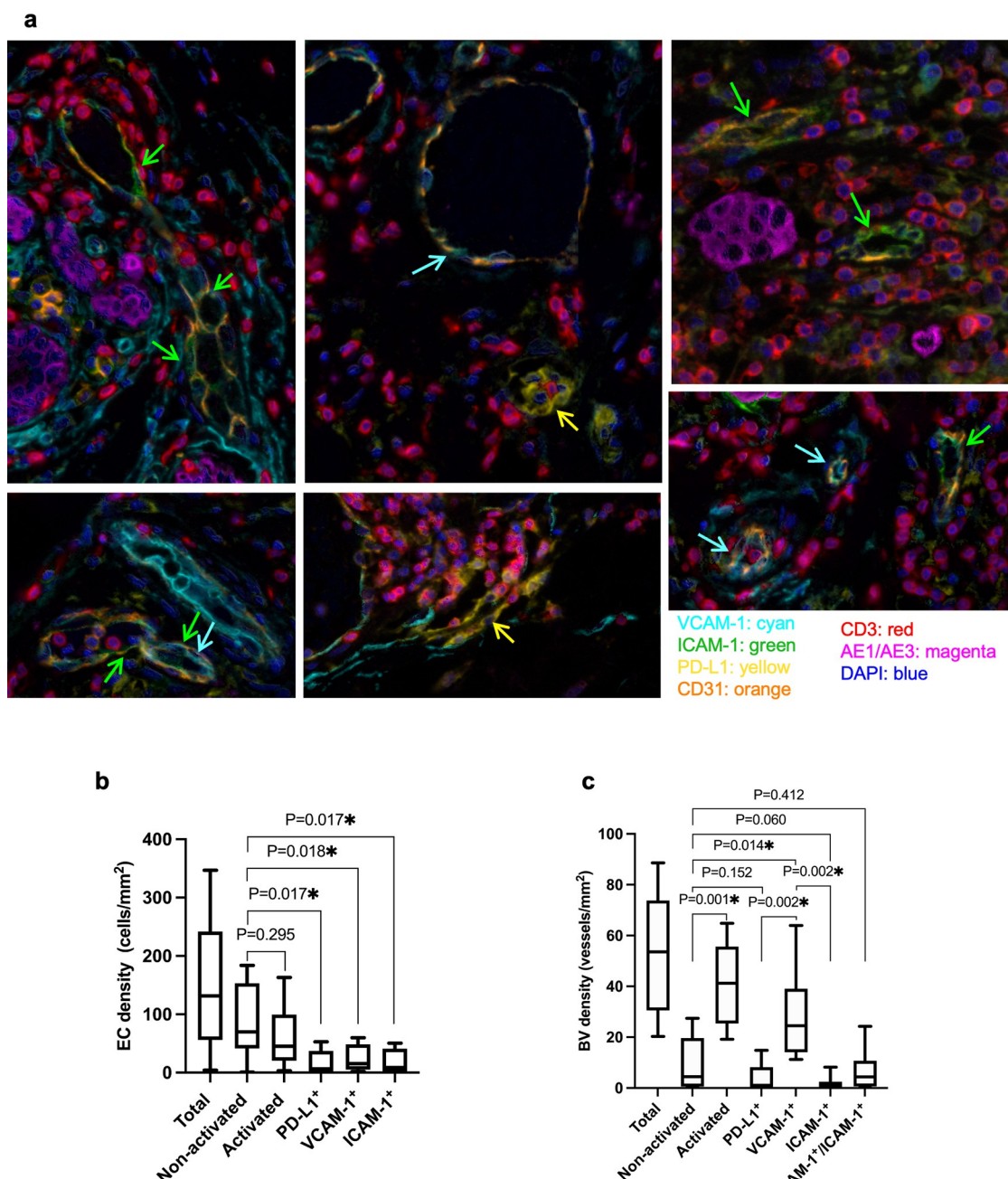

**Fig 3. The examples of activated (adhesion molecule positive) BVs and their densities (BVs and ECs) in 10 basal TNBC samples. a**) Examples of BV endothelium demonstrating the different phenotypes (VCAM-1[+] vessels indicated with cyan arrows, ICAM-1[+] vessels with green arrows and PD-L1[+] vessels with yellow arrows). **b**) EC density by each phenotype (total activated, activated PD-L1, activated VCAM-1, activated ICAM-1 and non-activated) and **c**) BV densities by these phenotypes.

or in non-activated BVs (P = 0.280) (Fig 5). This disparity suggests a distinctive T cell gradient exclusive to ICAM-1[+] BVs (P = 0.028), which was evidently absent in any other types of activated or non-activated BVs (see supporting information, S5 Fig).

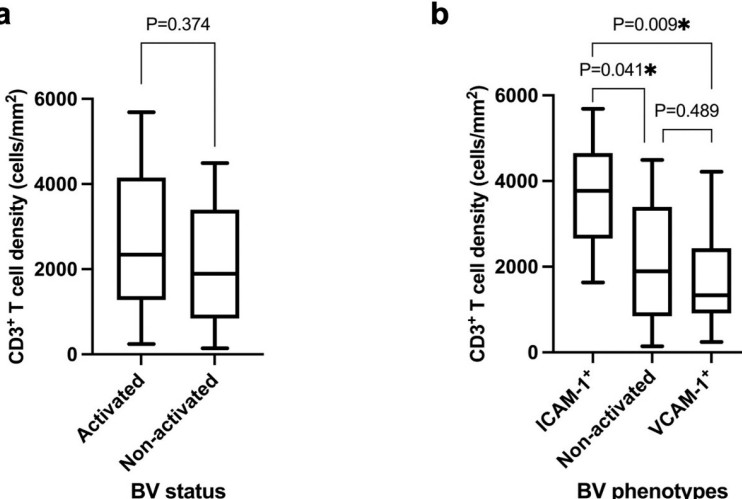

**Fig 4. T cell density within the 30μm zone of BVs. a)** T cell densities within 30μm zone of BVs were not significantly different between activated BVs (all phenotypes combined) and non-activated BVs. **b)** A significant difference in the CD3+ T cell density was evident in the 30μm zone of BVs for ICAM-1 activated BVs compared with non-activated BVs (P = 0.041) and VCAM-1 activated BVs (P = 0.009). There was no significant difference in CD3+ T cell densities between VCAM-1 and non-activated BVs (P = 0.489).

## *In Silico* analysis

Significantly higher TILs were observed in basal breast cancers compared with luminal A (P = 1.17 x $10^{-14}$) and luminal B (P = 4.41 x $10^{-5}$) with equivocal significant difference in the HER2 amplified subtype (P = 0.056, Fig 6A). There was a notable upregulation of the gene set used in the analysis (gene expression in immune cells in the tumour microenvironment and genes involved in the endothelial activation/adhesion) especially in the basal breast tumours that had higher numbers of TILs (Fig 6B, upregulated cluster). Indeed, the basal breast tumours were significantly associated with the "upregulated" cluster as well as indicating a low

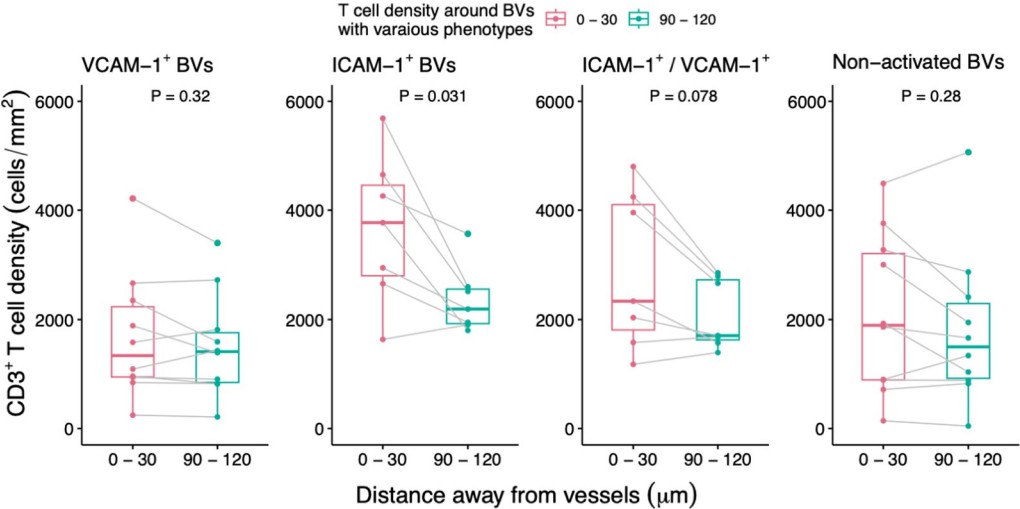

**Fig 5. CD3+ T cell densities in the most proximal zone (0-30μm) compared with the most distal zone (90-120μm) from BVs.** Significantly higher CD3+ T cell densities were observed around the 30μm zone of ICAM-1+ BVs compared with the distal 30μm zone at 90μm -120μm (P = 0.031)(paired two-tailed Wilcoxon rank-sum test).

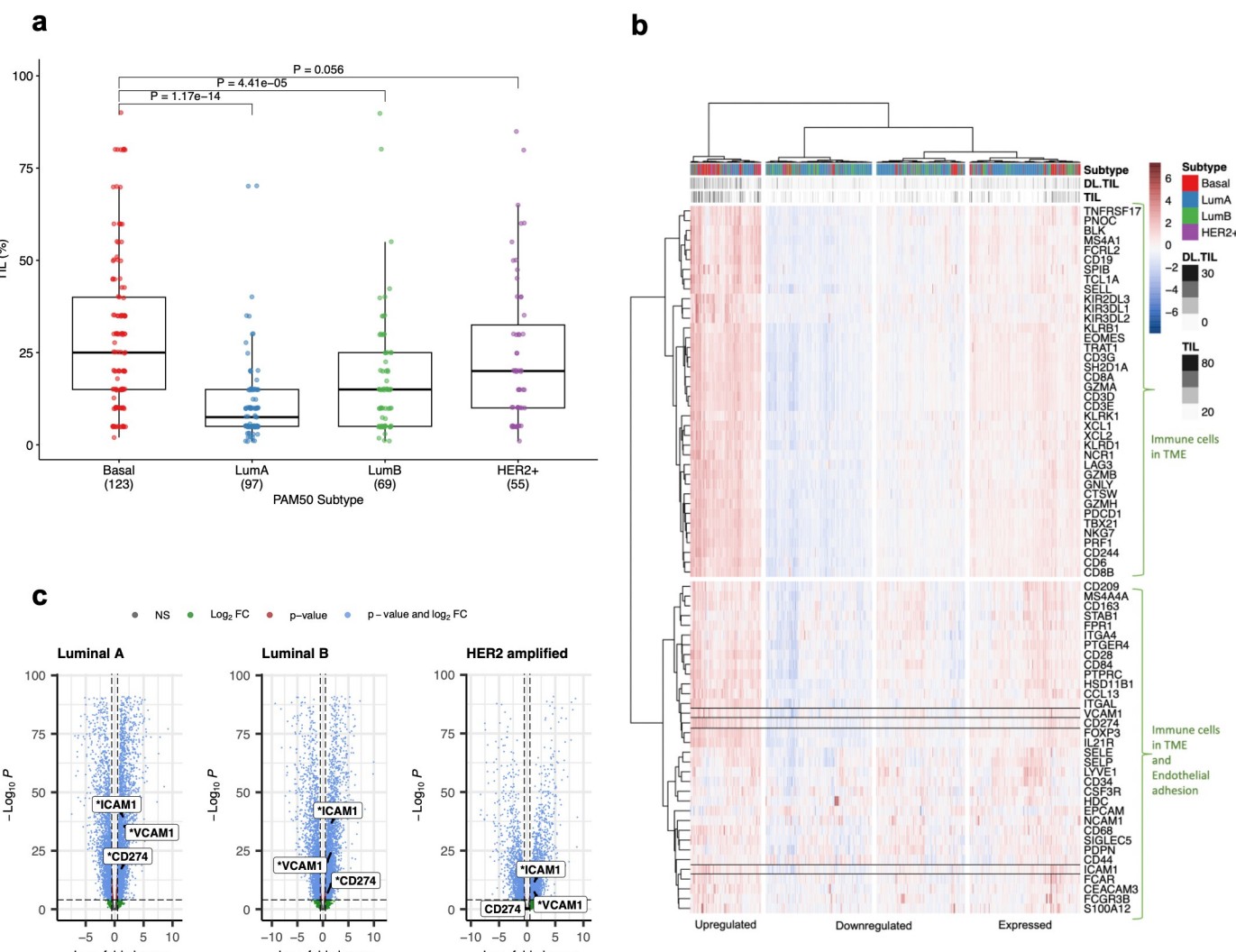

**Fig 6. In silico analysis of tumour microenvironment using the TCGA breast cancer data set. a) The summary of the TCGA cohort illustrates the range of TILs across different subtypes, along with the corresponding number of cases within each subtype (indicated in the brackets).** There are significantly higher TILs in the basal type compared with luminal A (LumA) and luminal B (LumB) but not with HER2+ subtypes (a two-sided Wilcoxon ranked-sum test adjusted for false discovery rate). **b). Hierarchical clustering data analysis of immune regulated genes and vascular endothelial adhesion molecules segmented for cells associated with the microenvironment in TCGA Provisional Breast Invasive Carcinoma Panel.** Heatmap of standardised variance-stabilising transformed counts, clustered by rows and by columns using a hierarchical clustering method with Euclidean distance and Ward's agglomeration. Each row represents a gene from the 72-gene set comprised of immune cells in the tumour microenvironment and activation adhesion molecules of interest. Each column represents a case (N = 808). The red color indicates upregulation, while blue indicates downregulation. Column annotation shows the PAM50 subtype and the two available TIL measurements: DL.TIL (deep learning TILs) and TIL (scored by a pathologist). The column dendrogram is divided into 4 clusters based on standardised values, categorised as Upregulated, Downregulated, and Expressed. The row dendrogram is divided into 2 clusters, with *ICAM-1*, *VCAM-1*, and *PD-L1 (CD274)* in the bottom cluster. **c) Differential gene expression analysis by DESeq2 methodology. Each volcano plot compares basal subtype vs the other subtype (LumA, LumB or HER2).** *ICAM-1* and *VCAM-1* are significantly upregulated in basal subtype compared with all the other subtypes. *PD-L1* (CD274) expression is significantly upregulated in basal compare with Luminal A and B but not significantly different in HER2 subtype. The positive log fold change indicates gene upregulation in basal compared with the other subtype. P-values are adjusted for the false discovery rate, with a significance level of α = 0.01. * indicates significance.

association with the "downregulated" cluster (see supporting information, S6 Fig). Pearson Residuals for Association, $\chi^2$ = 71.40; df = 6; p = 2.1 x $10^{-13}$). Furthermore, the differential expression confirmed that the endothelial activation markers of interest, *ICAM-1*, *VCAM-1* showed significantly higher expression in the basal subtype compared with all other subtypes and with *PD-L1* in basal compared with luminal A and B but not with HER2 subtypes (Fig 6C).

## Discussion

In this study, we have spatially characterised the tumour microenvironment in basal breast cancers to increase the understanding of the important relationship between tumour infiltrating lymphocytes and the blood vasculature. Since most of the TILs in breast cancer are made up of T cells [28] and this cell type is also the main effecter arm of the anti-cancer immune response, we specifically investigated the relationship between T cell density and the activation status of vascular endothelium. Our aims were to better understand the process of TILs trafficking into tumour microenvironment and to identifying potential targets and mechanisms that could potentially be modulated in so called TILs low "cold tumours" to improve their likelihood of responding to immune therapies. Basal TNBC rather than other breast cancer subtypes was a chosen for study due to its strong association with the immune response having TILs, a better clinical outcome in patients with high TIL tumours and their more frequent response to immunotherapy [10, 29, 30].

In this study, our *in silico* analysis confirmed that T cells are significantly more prominent in basal subtype compared with luminal A and B but not with HER2 and our spatial analysis showed that when present, CD3$^+$ T immune cells mostly reside in the stroma. To understand further how these CD3$^+$ T cells are trafficked into the stroma, we also studied tumour associated BVs with the expression of the key molecules VCAM-1 and ICAM-1 that are involved in immune cell trafficking.

We observed that in tumours, over eighty percent of the BVs are activated, which intuitively suggests a mechanism for immune cell trafficking into the tumour microenvironment. However, in contrast to the anticipated positive correlation between EC/BV density and CD3$^+$ T cells we observed no such relationship. Nevertheless, when examining spatial relationship between BVs and CD3$^+$ T cells enabled by the HALO software to delineate precise distances between these structures and cells, we observed a gradient of a differential density of CD3$^+$ T cells around BVs proximally on ICAM-1 activated vessels that was not present on VCAM-1or VCAM-1/ICAM-1 activated BVs or non-activated BVs.

VCAM-1 and ICAM-1 play distinct roles in the trans-endothelial migration cascade during immune cell trafficking. The trans-endothelial migration of leukocytes is a multistep process involving sequential activation of various cell adhesion molecules on ECs lining BVs and their cognate ligands on leukocytes. Briefly, the initial part of the cascade comprises T cell arrest and adhesion which are performed by both VCAM-1 and ICAM-1, however only ICAM-1 facilitates the active migration of the adhered leukocytes across the endothelium via either a transcellular or paracellular mechanism controlled by cytokines and chemokines [17, 31]. The VCAM-1 protein is subjected to degradation by TRIM65 E3 ligase, which does not affect ICAM-1 abundance [32]. Hence, this is likely accounts for the significant association of high CD3$^+$ TILs with ICAM-1 but not VCAM-1 activated BVs.

Our findings from the basal TNBC cases spatially analysed at the cellular level not only help to explain the observations from the *in silico* analysis of the TCGA breast cancer dataset but also provides insights into their implications at the tumour level. The analysis of the dataset revealed a strong association between the basal subtype and elevated TILs. These basal tumors exhibited higher expression of genes typically associated with TILs (e.g. *PTPRC*, *FOXP3* and *ITGAL* etc), affirming the high TILs presence in these cases. Moreover, they demonstrated higher expression levels of *ICAM-1* and *VCAM-1*, suggesting a mechanism of enhanced adhesion to the activated endothelium that is involved in the trans-endothelial migration of TILs, particularly in the presence of cytokines and chemokines, as indicated by the elevated expression of *IL21R*, *CD209* and *CCL13*. Along with the elevated levels of *ICAM-1* and *VCAM-1* in

basal subtype, *PD-L1* (CD274) expression was also significantly upregulated in basal compare with Luminal A and B subtypes.

The PD-1/PD-L1 signaling pathway has been in the spotlight in the recent years as its blockade with checkpoint inhibitor therapy in several tumour types including TNBCs re-engages the immune response leading to prolonged survival [33–35]. In TNBCs, this is likely to occur only in the presence of a robust immune cell infiltration since PD-L1 expression has been reported to be mostly expressed by the immune cell rather than the neoplastic cell compartment. In this study we observed PD-L1 expression not only on immune cells but also on BV endothelium suggesting an additional immunoregulatory role in TNBCs. One potential mechanism of PD-L1 upregulation in the BVs of these tumours is via tumour hypoxia, a common event in TNBC [36–39]. Tumour hypoxia is recognized to stabilise HIF-1alpha resulting in overexpression of PD-L1 [40, 41]. These PD-L1 expressing BVs have the potential to foster an immunosuppressive tumour microenvironment by inducing regulatory T cells (Treg) [42–44], which we have previously shown to confer a poor outcome in breast cancer [45]. The hypoxic response is also recognized to upregulate VEGF-A which not only promotes angiogenesis and formation of new vessels that are abnormal in structure, that contributes to ischemia from poor blood flow and the subsequent the tumour hypoxic microenvironment [46], but also may further augment the immunosuppressive tumour microenvironment in TNBC as it also mediates T cell exhaustion by increasing PD-1 expression on CD8[+] T cells [41, 47]. These findings demonstrate the important of the endothelial cell compartment within the stroma on the immune response in TNBC [40, 48–51].

## Summary

In conclusion, we have shown the positive association between the activated status of the vessels and T cell densities in basal TNBCs in a gradient dependent manner, findings supported by data from the *in silico* analysis. We noted that PD-L1 expression on ECs suggests an immune regulatory role. This raises the possibility that anti-angiogenic therapies may augment immunotherapies by normalising the aberrant tumour vasculature structure thereby reducing the hypoxic drive and by regulating the cell adhesion molecules and checkpoint molecule status of the vascular endothelium and immune cells. Models to induce 'immune cold' TNBCs to "hot in this difficult to treat population of patients might further elucidate the link between activation and immunomodulatory molecules on BVs and immune cells. It will be important to validate these findings in an independent cohort.

## Supporting information

**S1 Fig. Tissue segmentation using HALO software. a) and d)** tumour compartment defined by the positive expression of the pan-cytokeratin marker visualised by DAB, **b) and e)** the corresponding same tumour compartment (red) together with stroma (green) and other (blue) defined by HALO software, **c) and f)** merged mIF images.
(PDF)

**S2 Fig. Specific cell phenotyping using HALO software.** Cell phenotype showing in the 'real-time view window' (the square frame). The real-time view window and HALO software confirms as blue the DAPI signal in each nucleus from the raw IF image, and again the HALO software applies different coloured rings around nuclei, delineating the multiplex co-localisation of multiple markers and enabling specific cell phenotyping.
(PDF)

**S3 Fig. Zonal compartmentalisation for T cell infiltration analysis around BVs using HALO software.** CD3$^+$ density (cells/mm$^2$) up to 120μm from each blood vessel consisting of 4 bands of 30um indicated by the coloured bands (red, yellow, green, blue), is measured collectively for each BV phenotype using the HALO Spatial Analysis module.
(PDF)

**S4 Fig. CD3$^+$ T cells in tumour compartment and stromal compartment displaying their activation markers.**
(PDF)

**S5 Fig. CD3$^+$ T cell densities shown in each 30μm-width zone with 30μm increments; 0-30μm, 30-60μm, 60-90μm, 90-120μm from BVs, revealing high to low T cell gradient around ICAM-1$^+$ BVs from proximal to distal.** The distinctive T cell gradient exclusive to ICAM-1$^+$ BVs, which was evidently absent in any other types of activated or non-activated BVs. It was supported by a significant negative correlation (Spearman correlation R = -0.42, P = 0.028). R specifies the Spearman correlation, the line indicates the regression line and the grey shaded area specifies standard error in the graphs.
(PDF)

**S6 Fig. Statistical analysis showing the significant association between basal subtype and 'upregulated' gene cluster comprising genes involved immune regulation and endothelial activation/adhesion.** Depiction of Pearson residuals ($r^2$) by cluster and PAM50 subtype, derived from the Chi-square test of independence. The size of the circle is proportional to the residual's contribution. Red indicates a positive association, and blue indicates a negative association.
(PDF)

**S1 Table. BV quantification data.** Showing the BVs quantified in each phenotype in a single sample as an example, using HALO AI software (supporting data for Fig 3C).
(XLSX)

**S1 File. Supporting information on mIF, and segmentation of tumour compartment and cell types.**
(DOCX)

## Acknowledgments

We thank Judy Browning and Anatomical Pathology Department for their assistance in conducting IHC on Ventana BenchMark Ultra. David Byrne for assistance in preliminary TCGA data visualization. Centre for Advanced Histology and Microscopy core facility at Peter Mac in particular Cameron Skinner and Marné Prinsloo for their technical assistance with Vectra 3 Imaging System and HALO Image Analysis Software and the support provided by the Australian Cancer Research Foundation for the facility.

## Author Contributions

**Conceptualization:** Stephen B. Fox.

**Formal analysis:** Elena A. Takano, Luis E. Lara Gonzalez, Roberto Salgado.

**Funding acquisition:** Stephen B. Fox.

**Investigation:** Elena A. Takano, Jia-Min B. Pang.

**Methodology:** Elena A. Takano, Metta K. Jana, Luis E. Lara Gonzalez.

**Project administration:** Elena A. Takano.

**Resources:** Elena A. Takano, Jia-Min B. Pang, Sherene Loi.

**Software:** Elena A. Takano, Metta K. Jana, Luis E. Lara Gonzalez.

**Supervision:** Stephen B. Fox.

**Validation:** Elena A. Takano.

**Visualization:** Elena A. Takano, Stephen B. Fox.

**Writing – original draft:** Elena A. Takano.

**Writing – review & editing:** Elena A. Takano, Luis E. Lara Gonzalez, Jia-Min B. Pang, Stephen B. Fox.

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
