## [Decision Letter · Decision Letter 0]

7 Aug 2024

PONE-D-24-17184Characterising the spatial immune and vascular environment of triple negative breast carcinomas by multiplex fluorescent immunohistochemistry.PLOS ONE

Dear Dr. Takano,

Thank you for submitting your manuscript to PLOS ONE. After careful consideration, we feel that it has merit but does not fully meet PLOS ONE’s publication criteria as it currently stands. Therefore, we invite you to submit a revised version of the manuscript that addresses the points raised during the review process.

We look forward to receiving your revised manuscript.

Kind regards,

Yasuhiro Miki

Academic Editor

PLOS ONE

Journal Requirements:

"SBF is supported by NHMRC Investigator grant GNT1193630. (https://www.nhmrc.gov.au/funding/find-funding/investigator-grants)"

Reviewers' comments:

Reviewer's Responses to Questions

**Comments to the Author**

1. Is the manuscript technically sound, and do the data support the conclusions?

Reviewer #1: Partly

Reviewer #2: Partly

2. Has the statistical analysis been performed appropriately and rigorously? 

Reviewer #1: I Don't Know

Reviewer #2: Yes

3. Have the authors made all data underlying the findings in their manuscript fully available?

Reviewer #1: No

Reviewer #2: Yes

4. Is the manuscript presented in an intelligible fashion and written in standard English?

Reviewer #1: Yes

Reviewer #2: Yes

5. Review Comments to the Author

Reviewer #1: Dear authors,

With great pleasure I have read your submission “Characterising the spatial immune and vascular environment of triple negative breast carcinomas by multiplex fluorescent immunohistochemistry”.

Interesting to know more about the vessels in TNBC, their activation status and association to CD3+ T cells. While reviewing the manuscript I had some major and minor concerns which I believe have to addressed during a major revision.

Comments to authors:

1. Is the manuscript technically sound, and do the data support the conclusions?

Partly. I feel that the manuscript lacks in supporting information of mainly example mIHC images and subsequent analysis.

2. Has the statistical analysis been performed appropriately and rigorously?

I am not sure, but often just a T-test has been used, also in graphs where multiple groups are compared.

3. Have the authors made all data underlying the findings in their manuscript fully available?

No. Not all imaging data has been made available or the downstream images thereof.

4. Is the manuscript presented in an intelligible fashion and written in standard English?

Yes, only few typos.

Major concerns:

- Line 111: The sample size of 10 TNBC concerns me a little bit. It does not seems to be a big cohort from which you can derive major conclusion.

- I did not know that basal breast cancer and TNBC are often used interchangeably. I read that “Most but not all of basal-like breast cancers are triple-negative breast cancers and vice versa”. Specify this somewhere.

- figure 1: I miss representative images of the mIHC stainings. There are a few in the supplemental, but add some to the main figure.

- line 372: CD3 T cells were not quantified in the stroma of luminal A,B, HER2 and basal type. From this in sillico data you can not derive that.

- Also figures about how the analysis is performed would be a great addition to the paper. Tumor/stroma/background segmentation. The increment analysis. Detection of BVs.

Minor concerns:

- line 32: I am not sure if you can state that TNBC are highly immunogenic. There might be a higher TIL infiltrate then other breast cancers, but the TMB is not very high.

- line 43: mention all the markers used for the mIHC panel

- line 64: consider “and upregulated HER2 expression” and leave out “the latter as defined….”

- line 70: again, I would not state that TNBC is highly immunogenic, but rather are considered more “hot” tumor due to the TIL infiltrate.

- line 93: abbreviation of “TEM” is used only once in introduction and further only used twice in discussion. Consider to not use the abbreviation.

- line 127: AE1/AE3 is the clone name for the antibody that stains for pan-cytokeratin.

- Line 127: nothing is mentioned that the mIHC had to be optimized.

- Table 1: information about catalogue number and company is missing from primary antibodies.

- line 136: the use of Phenochart to select the tissue is not mention. How did the selection of 20x images take place? Focused on the tumor area and a few images into the stroma, or just the whole tissue including fat?

- line 140: it would be nice to include some images from examples of the tissue segmentation. At least mention the tissue categories defined here. Did you also include a category for background?

- line 148: show examples of cell quantification

- line 156: show examples of BV quantification and their positivity for the markers

- line 165: also mention here what has been mentioned in line 284

- line 176: mention that basal subtype corresponds to the TNBC.

- line 196: isn’t it weird that there are 42,119 genes analyzed while the human genome only contains about 20,000-25,000 genes (I’m not an expert in this though)?

- line 202: I read about T-test while in quite some figures you compare multiple groups to each other, which requires ANOVA as far as I know.

- line 211: a space is lacking between “range12.7-2419.8”

- line 214: you quantify the amount of T cells in tumor and stroma, but it would also be good to quantify the amount of BV in tumor and stroma if possible. Also a schematic overview of the distribution of 1 or more whole slides would be interesting to learn more about the distribution of BVs.

- line 222: no examples of CD3+ T cells are shown to express markers of VCAM-1, ICAM-1 and PD-L1.

- line 224: I miss an introductory sentence about why you would want to analyse the expression of these markers on T cells.

- line 237: in my experience it is hard to determine if the PD-L1 is really expressed on the CD3+ T cell itself, or if it is just expressed by the surrounding of the CD3+ T cells. How confident are you that you take the expression of all of these markers on the T cell and not just of its surrounding.

- line 243: show example in supplementary.

- line 247: I really like the fact that BVs that express ICAM-1 and VCAM-1 are shown in supplementary (consider to put in the main figure). I would also like to see PD-L1+ BV in an example.

- line 250: are the ECs only quantified for the tumour compartment and not for the stromal? This is not clear.

- line 265: are the BVs only quantified for the tumour compartment and not for the stromal? This is not clear.

- line 280: mentioning of these examples in fig s1 comes very late. This should be included in one of the main figures.

- line 284: show examples how HALO determine these increments. Is background category here excluded? What about neighboring ICAM-1+ BVs, are these counted double if they overlap?

- line 285: mentions figure 3 but first results on figure 4a-b are discussed and then the results in figure 3 at the end. Consider renumbering for clarity.

- figure 3: how was the statistical testing done for this figure? It should be an ANOVA I believe.

- line 315: for me it was very confusing that there is a switch between TNBC to basal and other molecular subtypes which have not been used in the introduction.

- figure 5-8: consider this to be implemented in one main figure or one main figure and some supplemental figures.

- fig 8: resolution of this figure is quite bad. I can’t distinguish the red and black dots used in here. Also mention on top of these figures each time basal vs…

- line 379: use of “T cell cells” is a bit weird

- line 385: term TEM comes back. I would not abbreviate it, it is used too little to do that.

- line 391-392: I don’t understand this sentence, rephrase please “does not affect ICAM-1 phenotype abundance hence VCAM-1 but not ICAM-1”.

- line 413: is it likely that PD-L1 expression on BV is induced due to hypoxia? I suppose BVs themselves are not hypoxic at all.

Reviewer #2: The paper is well organized. The objective of this study was to examine the relationship between the activation status of blood vasculatures and immune infiltrates in the context of TNBC. Specifically, the authors aimed to characterize T cells, including any preferential spatial location in relation to the vasculature and the activation status of the endothelium. In addition, the expression of PD-L1 in epithelial, endothelial and stromal component of basal-like TNBCs was explored. The overarching goals were to gain a better understanding of the process of TILs trafficking into the tumor and to identify potential targets that could be modulated to improve the likelihood of responses to immune therapies.

However, I noticed that the authors cohort is composed of only 10 cases of a basal-like phenotype with positivity for EGFR and/or CK5/6 immunohistochemistry. The cases were selected to contain a range of TILs (range 5% - 90%, median 65%) as assessed by the method of the International Immuno-Oncology Biomarker Working group. The authors added in silico analysis of the TCGA breast cancer dataset, in this study they confirmed these T cells are significantly more prominent in basal subtype compared with luminal A and B but not with HER2 and that when present, CD3+ T immune cells mostly reside in the stroma.

In my opinion is important to assess with the same mIF panel other TNBC cases belonging to other molecular categories (luminal A, Luminal B, etc. The finding that PD-L1 expression not only on immune cells but also on BV endothelium suggesting an additional immunoregulatory role in TNBCs. Is a novel finding that needs to be further explored increasing the cohort numbers and including different molecular subgroups.

6. PLOS authors have the option to publish the peer review history of their article (what does this mean?). If published, this will include your full peer review and any attached files.

Reviewer #1: No

Reviewer #2: No

---

## [Author Response · Author response to Decision Letter 0]

20 Sep 2024

We would like to thank both reviewers for the time and effort in assessing the manuscript. We are greatly appreciative. Please refer to our response addressing each comment. 

Reviewer #1: 

Dear authors,

With great pleasure I have read your submission “Characterising the spatial immune and vascular environment of triple negative breast carcinomas by multiplex fluorescent immunohistochemistry”.

Interesting to know more about the vessels in TNBC, their activation status and association to CD3+ T cells. While reviewing the manuscript I had some major and minor concerns which I believe have to addressed during a major revision.

Comments to authors:

1. Is the manuscript technically sound, and do the data support the conclusions?

Partly. I feel that the manuscript lacks in supporting information of mainly example mIHC images and subsequent analysis.

We have addressed this point and included multiple additional images both in the main manuscript itself and also in the supplemental materials document for further reference. Please refer below for each of the extra figures and images we have added to address your comments. 

2. Has the statistical analysis been performed appropriately and rigorously?

I am not sure, but often just a T-test has been used, also in graphs where multiple groups are compared.

The approach for the data analysis was provided by an experienced bioinformatician/statistician who conducted appropriate statistical analyses including the correct tests for comparing multiple groups. 

3. Have the authors made all data underlying the findings in their manuscript fully available?

No. Not all imaging data has been made available or the downstream images thereof.

We have as above included additional images in the revised version. Although we could potentially upload all images we believe that the data referred to in the question pertains more to genomic/quantitative information rather than images and trust this is the case. Nevertheless, we would be happy to upload images on advice from the Editor. 

4. Is the manuscript presented in an intelligible fashion and written in standard English?

Yes, only few typos.

We have corrected the typos.

Major concerns:

- Line 111: The sample size of 10 TNBC concerns me a little bit. It does not seems to be a big cohort from which you can derive major conclusion.

The design of the study was to use a highly selected discovery cohort that had been chosen to cover specific ranges of TILs within which then could be validated in a larger in silico independent cohort. The TNBC had been previously screening from a large cohort to contain a range of TILs from low to high. This discovery cohort revealed relationships between CD3+T cells and vessels, an observation confirmed by the in silico analysis thus supporting our hypotheses. The only rationale for using a larger cohort would be if no relationships was observed. We of the opinion that study design with additional strong in silico analysis are sufficient to support the findings in the study. 

- I did not know that basal breast cancer and TNBC are often used interchangeably. I read that “Most but not all of basal-like breast cancers are triple-negative breast cancers and vice versa”. Specify this somewhere.

We have clarified the relationship for the reader by adding in the manuscript text (lines 82-84 in the revised version) in Introduction “TNBC is a heterogeneous group of tumours that can be classified further into subtypes, the most common being basal-like, based on their morphology and transcriptomic profile (ref, WHO classification of Tumours – Breast Tumours).” We hope this makes it clear that we are focussed on the most frequent aggressive form of TNBC. Furthermore, we now use “TNBCs of basal type” “basal type TNBC” and “basal TNBC” throughout the manuscript for clarity. 

- figure 1: I miss representative images of the mIHC stainings. There are a few in the supplemental, but add some to the main figure.

We have added images (Fig 2 and 3) to the Results section in the revised version, specifically in the sub-sections “A subset of tumour-associated CD3+ immune cells…” and “A subset of ECs and BVs in TNBC express…”.

- line 372: CD3 T cells were not quantified in the stroma of luminal A,B, HER2 and basal type. From this in sillico data you can not derive that.

We agree that this is not clear. We have clarified this statement conclusions come from the spatial analysis on 10 basal cases, and which are from in silico (whole tumour) analysis. “In this study, our in silico analysis confirmed that T cells are significantly more prominent in basal subtype compared with luminal A and B but not with HER2 and our spatial analysis showed that when present, CD3+ T immune cells mostly reside in the stroma.” (Discussion, 2nd paragraph line 589, revised version)

- Also figures about how the analysis is performed would be a great addition to the paper. Tumor/stroma/background segmentation. The increment analysis. Detection of BVs.

Please refer to the newly added figures (S1 Fig., S2 Fig. and S3 Fig.) in the supplementary data and in the supporting information that outlines in greater detail how this was performed.

Minor concerns:

- line 32: I am not sure if you can state that TNBC are highly immunogenic. There might be a higher TIL infiltrate then other breast cancers, but the TMB is not very high.

It is generally recognised that TNBC elicit a strong immune response in many patients (e.g. Front Genet 2022; 13: 1095839). Histologically this is frequently reflected in the presence of large numbers of TILs. 

Nevertheless, we have replaced “highly immunogenic” by “often contain higher numbers of TILs compared with other breast cancer subtypes” (1st line in the Abstract, i.e. line 32 in the revised version). 

- line 43: mention all the markers used for the mIHC panel

With the word count restriction in the abstract, we listed all the markers in the materials and methods section instead. However, we would be happy to include them in the abstract if the editors allow for the extra word count needed. 

- line 64: consider “and upregulated HER2 expression” and leave out “the latter as defined….”

Yes, “the latter” was deleted and the sentence now reads better, thank you. 

- line 70: again, I would not state that TNBC is highly immunogenic, but rather are considered more “hot” tumor due to the TIL infiltrate.

As above “highly immunogenic” has been deleted (Introduction, line 90 in the revised version).

- line 93: abbreviation of “TEM” is used only once in introduction and further only used twice in discussion. Consider to not use the abbreviation.

We have removed the abbreviation. 

- line 127: AE1/AE3 is the clone name for the antibody that stains for pan-cytokeratin.

We have changed this to “pan-cytokeratin” with the clone AE1/AE3 (Materials and Methods, Immunohistochemistry, line 160 in the revised version).

- Line 127: nothing is mentioned that the mIHC had to be optimized.

Staining conditions for each antibody were optimised. We have added “optimised” in Table 1 heading to emphasise the staining conditions used were optimised. Please refer to mIF details in the separate ‘supporting information’ which outlines the optimised staining conditions including the antibody staining order is listed in the Table 1. Also “including optimised staining conditions” is added in Materials and Methods section, under Immunohistochemistry, line 162 in the revised version. 

- Table 1: information about catalogue number and company is missing from primary antibodies.

The company name and catalogue numbers have been added in the manuscript main body under “Immunohistochemistry” sub-heading in Materials and Methods (lines 159-161 in the revised version).

- line 136: the use of Phenochart to select the tissue is not mention. How did the selection of 20x images take place? Focused on the tumor area and a few images into the stroma, or just the whole tissue including fat?

We have added the additional information to describe this (under “Image acquisition and segmentation of tumour compartments” lines 174- 178 in the revised version). “The ROIs were selected using a whole slide image viewer, Phenochart (Akoya Biosciences). Areas from the tumour periphery which is recognised to be most biologically important (APMIS 112: 413–30, 2004) and where the cells of our interest such as immune cells, microvasculature endothelial cells and tumour cells are most prominently located.”

- line 140: it would be nice to include some images from examples of the tissue segmentation. At least mention the tissue categories defined here. Did you also include a category for background?

Yes, the tissue segmentation classes include Tumour (red), Stroma (green) and Other/ background (blue). Please refer to the newly added S1 Fig. in supplementary document for how each category was defined, including the ‘other’ category.

- line 148: show examples of cell quantification

Please refer to the newly added S2 Fig. in supplementary document. Cells were quantified accordingly to their cell phenotypes. 

- line 156: show examples of BV quantification and their positivity for the markers

Please refer to Fig. 3 for the examples of BVs expressing various markers and S1 Table for the BV quantification data.

- line 165: also mention here what has been mentioned in line 284

Line 284-285 (indicated at the line 378 in the revised version) deleted.

- line 176: mention that basal subtype corresponds to the TNBC.

As addressed in the previous comment. The additional line in the introduction should clarify this.

- line 196: isn’t it weird that there are 42,119 genes analyzed while the human genome only contains about 20,000-25,000 genes (I’m not an expert in this though)?

Our apologies for this error. 42,119 is the number of probes. We have corrected this to 16,570 genes (Materials and Methods, sub-section “In Silico analysis of …”, line 247 in the revised version).

- line 202: I read about T-test while in quite some figures you compare multiple groups to each other, which requires ANOVA as far as I know.

The P value corrections can be undertaken in multiple ways including ANOVA. However, we have chosen the Benjamin-Hochberg technique as advised by our biostatistician as a more robust method. 

We have now adjusted P values in the figures 4, 6 and S4 Fig (Original Fig 3 has been split into the new Fig 6 and S4 Fig. to show their P values clearly). All other P values in the other figures have been adjusted previously and appropriately (by B-H methods). 

- line 211: a space is lacking between “range12.7-2419.8”

We have corrected this, thank you.

- line 214: you quantify the amount of T cells in tumor and stroma, but it would also be good to quantify the amount of BV in tumor and stroma if possible. 

BVs by definition are always in stroma as they are variably surrounded by basement membrane, pericytes and stroma and are not in the tumour compartment. There is a concept, tumour mimicry, where tumour themselves make up the vasculature however this has not been shown to occur to any great extent in breast cancer and certainly not in our cases as this would have been shown by the cytokeratin staining. 

Also a schematic overview of the distribution of 1 or more whole slides would be interesting to learn more about the distribution of BVs.

The focus of the paper is to investigate the relationships between the immune cells and the vasculature not the distribution of blood vessels in breast cancer. The study of this, angiogenesis has been well studied previously by ourselves and multiple other groups. We would be happy to provide our references on this in many tumour types on request.

- line 222: no examples of CD3+ T cells are shown to express markers of VCAM-1, ICAM-1 and PD-L1.

Please refer to the new figure (Fig. 2a,b,c)

- line 224: I miss an introductory sentence about why you would want to analyse the expression of these markers on T cells.

The following line added (Discussion, lines 560- 562 in the revised version). “Since most of the TILs in breast cancer are made up of T cells (ref- Nat Rev Clin Oncol. 2016 Apr;13(4):228-41) and this cell type is also the main effecter arm of the anti-cancer immune response” we focused on this immune cell type. 

- line 237: in my experience it is hard to determine if the PD-L1 is really expressed on the CD3+ T cell itself, or if it is just expressed by the surrounding of the CD3+ T cells. 

We agree it can be challenging ensuring that there is true colocation of epitopes using conventional chromogenic immunostaining, largely due to resolution, especially determining the precise cell expressing this immune checkpoint marker. However, the ability to multiplex these using high resolution fluorescent markers provides strong data on co-expression on individual cells confirming the precise cellular phenotypes for multiple biomarkers including PD-L1.

How confident are you that you take the expression of all of these markers on the T cell and not just of its surrounding.

This is enabled using highly quantitative measurement performed by HALO using DAPI expression to identify cell nuclei and using serial additional biomarkers whether in the nucleus, cytoplasm or membrane to further refine the cell type. PD-L1 expression was identified in cytoplasm/membrane and appears outside of nuclei. By carefully adjusting cell segmentation and restricting the maximum cytoplasmic radius to 1�m in the cytoplasmic detection setting ensures accurate cell phenotyping. Other biomarkers are similarly adjusted to ensure correct cell typing.

- line 243: show example in supplementary.

All phenotypes of CD3+ T cells are collated, and the images have been added in Fig. 2.

d) CD3+/PD-L1+/VCAM-1+, e) CD3+/PD-L1+/ICAM-1+ and f)CD3+/PD-L1+/VCAM-1+/ICAM-1+

- line 247: I really like the fact that BVs that express ICAM-1 and VCAM-1 are shown in supplementary (consider to put in the main figure). I would also like to see PD-L1+ BV in an example.

The figure in the supplementary has been brought into the main manuscript. Images of PD-L1+BVs have been added to the figure (Fig 3.)

- line 250: are the ECs only quantified for the tumour compartment and not for the stromal? This is not clear.

As with the previous comment above on BVs, ECs making up BVs are only located in the stroma and not in the tumour compartment. 

The terminology ‘tumour’ has been used to mean overall tumour/ROI, not the tumour compartment. Both EC and BV density identified by positive CD31 expression were quantified “in the whole tumour (whole ROI) was added to clarify this (under sub-section “A subset of ECs and BVs…” line 319 in the revised version)

- line 265: are the BVs only quantified for the tumour compartment and not for the stromal? This is not clear.

As with the previous comment on BVs, they are in stroma and not in the tumour compartment. 

The terminology ‘tumour’ was used meaning overall tumour/ROI, not the tumour compartment. “Overall” was added to clarify this (under sub-section “A subset of ECs and BVs…” line 346 in the revised version). 

- line 280: mentioning of these examples in fig s1 comes very late. This should be included in one of the main figures.

The figure has been moved from supplementary to the main manuscript (now Fig. 3) and mentioned in the first line of the sub-section “A subset of ECs and BVs…” under Results (line 318 in the revised version) to address this concern. 

- line 284: show examples how HALO determine these increments. Is background category here excluded? What about neighboring ICAM-1+ BVs, are these counted double if they overlap?

The reviewer is correct to identify adjacent vessels as a possible confounder. This is precisely why BVs were only analysed where there where there were no other BVs nearby ensuring any relationship would not be contaminated by such an overlap. Furthermore, we investigated the most proximal zone (0-30�m) separately (Fig. 5) ensuring no influences from other vessels. The density of each zone/band was determined collectively for each vessel phenotype to avoid double ups in the T cell quantification (see Supplementary S3 Fig.). HALO software enables precise banding of zones of defined dis

---

## [Decision Letter · Decision Letter 1]

2 Dec 2024

PONE-D-24-17184R1Characterising the spatial immune and vascular environment of triple negative basal breast carcinomas by multiplex fluorescent immunohistochemistry.PLOS ONE

Dear Dr. Takano,

Thank you for submitting your manuscript to PLOS ONE. After careful consideration, we feel that it has merit but does not fully meet PLOS ONE’s publication criteria as it currently stands. Therefore, we invite you to submit a revised version of the manuscript that addresses the points raised during the review process.

We look forward to receiving your revised manuscript.

Kind regards,

Lu Zhang

Academic Editor

PLOS ONE

Journal Requirements:

Reviewers' comments:

Reviewer's Responses to Questions

**Comments to the Author**

1. If the authors have adequately addressed your comments raised in a previous round of review and you feel that this manuscript is now acceptable for publication, you may indicate that here to bypass the “Comments to the Author” section, enter your conflict of interest statement in the “Confidential to Editor” section, and submit your "Accept" recommendation.

Reviewer #1: (No Response)

Reviewer #2: All comments have been addressed

2. Is the manuscript technically sound, and do the data support the conclusions?

Reviewer #1: Yes

Reviewer #2: Partly

3. Has the statistical analysis been performed appropriately and rigorously? 

Reviewer #1: Yes

Reviewer #2: Yes

4. Have the authors made all data underlying the findings in their manuscript fully available?

Reviewer #1: No

Reviewer #2: Yes

5. Is the manuscript presented in an intelligible fashion and written in standard English?

Reviewer #1: Yes

Reviewer #2: Yes

6. Review Comments to the Author

Reviewer #1: Dear authors,

Thank you for addressing my comments so extensively.

I think this has taken the manuscript to a higher level.

Especially the incorporation of images in the manuscript makes it much more insightful and appealing.

I only have some minor comments left:

Line 69: now all other subtypes have been removed, but they are for importance in figure 7. I think it would help the reader if you at least mention the other subtypes of TNBC here as well.

Line 78-81: here is is stated that PD-L1 expression in TNBC is mostly on the tumor-infiltrating lymphocyte (TIL) compartment. This is an incorrect statement. When reviewing reference 13 (ref 14 may be removed because it is a podcast) it is stated that “PD-L1 expression on tumor-infiltrating immune cells”. Tumor-infiltrating immune cells and lymphocytes are not terms that are mutually exclusive. In general, PD-L1 is scored on either the tumor compartment or immune cell compartment. You can’t determine if the PD-L1 is on lymphocytes or on other immune cells. Please adjust this.

Line 223: It would be nice if this sentence refers to a new figure 1a-b where there is a multiplex image of T cells in the tumor and stromal compartment and then continue on for the next subnumbers with the quantification.

Figure 1: I noticed that there has not yet been made a nicely fitting lay-out for figure 1. I am not sure if this is conform PLOS ONE policy. Figure 1a, b, c and d are uploaded separately and not as one larger figure 1. It would have been nicer for the reviewing process to also be able to comment on this part.

Figure 2: The resolution of these images is quite bad again. There is also sort of a white box in the image, which is not clearly visible. It would be more clearly if next to these composite images, also black and white images are shown of the cells of interest as has been done in this publication (/doi.org/10.1002/eji.202350616) in figure 3.

Figure 3: also see if you can represent the co-expression in the example that is given for figure 2. Next to that, the examples of figure 3 can be merged with it quantification of figure 4.

Figure 7: resolution is quite bad.

Reviewer #2: Dear authors thanks for your response. While data from 10 cases can be valuable for exploratory purposes or hypothesis generation, the authors should frame their conclusions as preliminary and avoid making broad claims. I would recommend emphasizing the need for further studies with larger sample sizes to validate these results and strengthen the robustness of the findings. I suggest modifying the title as "Preliminary Characterization of the Spatial Immune and Vascular Microenvironment in Triple Negative Basal Breast Carcinoma Using Multiplex Fluorescent Immunohistochemistry"

7. PLOS authors have the option to publish the peer review history of their article (what does this mean?). If published, this will include your full peer review and any attached files.

Reviewer #1: **Yes: **Dr. Mark A.J. Gorris

Reviewer #2: No

---

## [Author Response · Author response to Decision Letter 1]

18 Dec 2024

We would like to thank both reviewers once again for their time and effort in evaluating the manuscript. We are deeply appreciative. Please refer to our responses below each comment, addressing each one.

Reviewer's Responses to Questions 

Comments to the Authors:

1. If the authors have adequately addressed your comments raised in a previous round of review and you feel that this manuscript is now acceptable for publication, you may indicate that here to bypass the “Comments to the Author” section, enter your conflict of interest statement in the “Confidential to Editor” section, and submit your "Accept" recommendation.

Reviewer #1: (No Response)

Reviewer #2: All comments have been addressed

Thank you.

2. Is the manuscript technically sound, and do the data support the conclusions?

Reviewer #1: Yes

Reviewer #2: Partly

The design of the study was to use a highly selected discovery cohort that had been chosen to cover specific ranges of immune infiltration within which then could be validated in a larger independent in silico cohort. This discovery cohort revealed relationships between CD3+T cells and vessels, an observation confirmed by the in silico analysis thus supporting our hypotheses. We of the opinion that study design with additional strong in silico analysis are sufficient to support the findings in the study. 

3. Has the statistical analysis been performed appropriately and rigorously? 

Reviewer #1: Yes

Reviewer #2: Yes

Thank you.

4. Have the authors made all data underlying the findings in their manuscript fully available?

Reviewer #1: No

All of our graphs are Box and Whisker Plots and visually displaying the data points distribution through their quartiles with upper and lower extremes. We also included representative images throughout in the manuscript. Although we could potentially upload all the scanned mIF images, we believe that the data referred to in the question pertains more to genomic/quantitative information rather than images and trust this is the case. Nevertheless, we would be happy to upload images on advice from the Editor. Please let us know how we can address this. Thank you.

Reviewer #2: Yes

5. Is the manuscript presented in an intelligible fashion and written in standard English?

Reviewer #1: Yes

Reviewer #2: Yes

Thank you.

6. Review Comments to the Author

Reviewer #1: 

Dear authors,

Thank you for addressing my comments so extensively. I think this has taken the manuscript to a higher level. Especially the incorporation of images in the manuscript makes it much more insightful and appealing.

Thank you, we agree that the paper is much stronger.

I only have some minor comments left:

Line 69: now all other subtypes have been removed, but they are for importance in figure 7. I think it would help the reader if you at least mention the other subtypes of TNBC here as well.

Thank you for your suggestion. We added the other subgroups and now it reads “TNBC is a heterogeneous group of tumours that can be classified further into subtypes based on their morphology and transcriptomic profile [2]. The most common is basal-like with other groups being claudin-low, mesenchymal, luminal androgen receptor and immunomodulatory.”

Line 78-81: here is is stated that PD-L1 expression in TNBC is mostly on the tumor-infiltrating lymphocyte (TIL) compartment. This is an incorrect statement. When reviewing reference 13 (ref 14 may be removed because it is a podcast) it is stated that “PD-L1 expression on tumor-infiltrating immune cells”. Tumor-infiltrating immune cells and lymphocytes are not terms that are mutually exclusive. In general, PD-L1 is scored on either the tumor compartment or immune cell compartment. You can’t determine if the PD-L1 is on lymphocytes or on other immune cells. Please adjust this.

We agree that immune cells are not synonymous with lymphocytes. However, most PD-L1 positive immune cells in TNBCs are lymphocytes and from personal scoring of hundreds of breast cancers for PD-L1, it is the lymphocytes that predominantly stain. Nevertheless, we have changed to use immune cells for ease.

Line 223: It would be nice if this sentence refers to a new figure 1a-b where there is a multiplex image of T cells in the tumor and stromal compartment and then continue on for the next subnumbers with the quantification.

We hope our understanding on your suggestion is correct. Line 223 in both versions of our revised manuscript (with track changes and clean version) belongs to the subsection “in silico analysis of tumour microenvironment on the TCGA breast cancer data” and “Statistical analysis” respectively. Hence we don’t see how these paragraphs we can relate to Fig 1. Nevertheless, we have interpreted this to add a figure, which we have done to supplementary information -S4.Fig (our preference) of CD3 T cells in tumour compartment and stromal compartment with the activated T cells in these compartments. 

Figure 1: I noticed that there has not yet been made a nicely fitting lay-out for figure 1. I am not sure if this is conform PLOS ONE policy. Figure 1a, b, c and d are uploaded separately and not as one larger figure 1. It would have been nicer for the reviewing process to also be able to comment on this part.

Apologies for not yet creating a multi-panel figure. This is because the figure preparation checklist states at the very top that "PLOS ONE waives all formatting requirements until your manuscript has received a provisional Editorial Acceptance decision.” However, we have now created a multi-panel figure. 

Figure 2: The resolution of these images is quite bad again. There is also sort of a white box in the image, which is not clearly visible. It would be more clearly if next to these composite images, also black and white images are shown of the cells of interest as has been done in this publication (/doi.org/10.1002/eji.202350616) in figure 3.

Apologies for the poor resolution. As recommended in the PLOS ONE checklist, we used the PACE tool to correct/reduce the file size before uploading. The white boxes around the cells of interest were generated by the HALO software. We acknowledge that the boxes appear faint however, they are designed to highlight the cell phenotypes without causing obstruction therefore, to aid in identifying the boxes, arrows were added as indicators. However, we have now made the boxes drawn with thicker lines. Also please refer to the new montages (bottom row), displaying a single channel with DAPI, for each of the corresponding panel d), e) and f).

Figure 3: also see if you can represent the co-expression in the example that is given for figure 2. 

We think co-expression of 2 markers on BVs (BVs are made up with multiple endothelial cells expressing different adhesion molecules) are clearly visible in this figure. 

Next to that, the examples of figure 3 can be merged with it quantification of figure 4. 

We have made a multi-panel figure, merging figure 3 with our quantification figure 4 as your suggestion. 

Figure 7: resolution is quite bad.

(Now Fig 6)

We apologize again for the poor resolution. However, we have followed the PLOS ONE figure guidelines and used the PACE tool for verification. Please refer to the newly created figure, which is now available in a larger file with improved resolution.

Reviewer #2: 

Dear authors thanks for your response. While data from 10 cases can be valuable for exploratory purposes or hypothesis generation, the authors should frame their conclusions as preliminary and avoid making broad claims. I would recommend emphasizing the need for further studies with larger sample sizes to validate these results and strengthen the robustness of the findings. I suggest modifying the title as "Preliminary Characterization of the Spatial Immune and Vascular Microenvironment in Triple Negative Basal Breast Carcinoma Using Multiplex Fluorescent Immunohistochemistry"

Thank you for your suggestion, we have changed the title as per your suggestion and have emphasised in the summary that validation of our findings is needed “It will be important to validate these findings in an independent cohort.”

---

## [Editor Report · Decision Letter 2]

27 Dec 2024

Preliminary Characterisation of the Spatial Immune and Vascular Microenvironment in Triple Negative Basal Breast Carcinoma Using Multiplex Fluorescent Immunohistochemistry

PONE-D-24-17184R2

Dear Dr. Elena A Takano,

We’re pleased to inform you that your manuscript has been judged scientifically suitable for publication and will be formally accepted for publication once it meets all outstanding technical requirements.

Kind regards,

Lu Zhang

Academic Editor

PLOS ONE
---

## [Editor Report · Acceptance letter]

30 Dec 2024

PONE-D-24-17184R2 

PLOS ONE

Dear Dr. Takano, 

I'm pleased to inform you that your manuscript has been deemed suitable for publication in PLOS ONE. Congratulations! Your manuscript is now being handed over to our production team.

Kind regards, 

on behalf of

Dr. Lu Zhang 

Academic Editor

PLOS ONE